# Learning Weighted Representations for Generalization Across Designs

## Abstract

Predictive models that generalize well under distributional shift are often desirable and sometimes crucial to machine learning applications. One example is the estimation of treatment effects from observational data, where a subtask is to predict the effect of a treatment on subjects that are systematically different from those who received the treatment in the data. A related kind of distributional shift appears in unsupervised domain adaptation, where we are tasked with generalizing to a distribution of inputs that is different from the one in which we observe labels. We pose both of these problems as prediction under a shift in *design*. Popular methods for overcoming distributional shift are often heuristic or rely on assumptions that are rarely true in practice, such as having a well-specified model or knowing the policy that gave rise to the observed data. Other methods are hindered by their need for a pre-specified metric for comparing observations, or by poor asymptotic properties. In this work, we devise a bound on the generalization error under design shift, based on integral probability metrics and sample re-weighting. We combine this idea with representation learning, generalizing and tightening existing results in this space. Finally, we propose an algorithmic framework inspired by our bound and verify is effectiveness in causal effect estimation.

## 1 Introduction

A long-term goal in artificial intelligence is for agents to learn how to *act*. This endeavor relies on accurately predicting and optimizing for the outcomes of actions, and fundamentally involves estimating counterfactuals—what would have happened if the agent acted differently? In many applications, such as the treatment of patients in hospitals, experimentation is infeasible or impractical, and we are forced to learn from *biased*, observational data. Doing so requires adjusting for the *distributional shift* between groups of patients that received different treatments. A related kind of distributional shift arises in unsupervised domain adaptation, the goal of which is to learn predictive models for a target domain, observing ground truth only in a source domain.

In this work, we pose both domain adaptation and treatment effect estimation as special cases of prediction across shifting *designs*, referring to changes in both action policy and feature domain. We separate policy from domain as we wish to make *causal* statements about the policy, but not about the domain. Learning from observational data to predict the counterfactual outcome under treatment $B$ for a patient who received treatment $A$, one must adjust for the fact that treatment $A$ was systematically given to patients of different characteristics from those who received treatment $B$. We call this *predicting under a shift in policy*. Furthermore, if all of our observational data comes from hospital $P$, but we wish to predict counterfactuals for patients in hospital $Q$, with a population that differs from $P$, an additional source of distributional shift is at play. We call this *a shift in domain*. Together, we refer to the combination of domain and policy as the *design*. The design for which we observe ground truth is called the *source*, and the design of interest the *target*.

The two most common approaches for addressing distributional shift are to learn shift-invariant representations of the data (Ajakan et al., 2014) or to perform sample re-weighting or matching (Shimodaira, 2000; Kallus, 2016). Representation learning approaches attempt to extract only information from the input that is invariant to a change in design *and* predictive of the variable of interest. Such representations are typically learned by fitting deep neural networks in which activations of deeper layers are regularized to be distributionally similar across designs (Ajakan et al., 2014; Long

et al., 2015). Although representation learning can be shown to reduce the error associated to distributional shift (Long et al., 2015) in some cases, standard approaches are biased, even in the limit of infinite data, as they penalize the use also of predictive information. In contrast, re-weighting methods correct for distributional shift by assigning higher weight to samples from the source design that are representative of the target design, often using importance sampling. This idea has been well studied in, for example, the causal inference (Rosenbaum & Rubin, 1983), domain adaptation (Shimodaira, 2000) and reinforcement learning (Precup et al., 2001) literature. For example, in causal effect estimation, importance sampling is equivalent to re-weighting units by the inverse probability of observed treatments (treatment propensity). Re-weighting with knowledge of importance sampling weights often leads to asymptotically unbiased estimators of the target outcome, but may suffer from high variance in finite samples (Swaminathan & Joachims, 2015).

A significant hurdle in applying re-weighting methods is that optimal weights are rarely known in practice. There are a variety of methods to learn these weights. Weights can be estimated as the inverse of estimated feature or treatment densities (Rosenbaum & Rubin, 1983; Freedman & Berk, 2008) but this plug-in approach can lead to highly unstable estimates. More stable methods learn weights by minimizing distributional distance metrics (Gretton et al., 2009; Kallus, 2016; 2017; Zubizarreta, 2015). Closely related, matching (Stuart, 2010) produces weights by finding units in the source design that are close in some metric to units in the target design. Specifying a distributional or unit-wise metric is challenging, especially if the input space is high-dimensional where no metric incorporating all features can ever be made small. This has inspired heuristics such as first performing variable selection and then finding a matching in the selected covariates.

Our key algorithmic contribution is to show how to combine the intuition behind shift-invariant representation learning and re-weighting methods by jointly learning a representation $\Phi$ of the input space and a weighting function $w(\Phi)$ to minimize a) the re-weighted empirical risk and b) a re-weighted measure of distributional shift between designs. This is useful also for the identity representation $\Phi(x) = x$, as it allows for principled control of the variance of estimators through regularization of the re-weighting function $w(x)$, mitigating the issues of exact importance sampling methods. Further, this allows us to evaluate $w$ on hold-out samples to select hyperparameters or do early stopping. Finally, letting $w$ depend on $\Phi$ alleviates the problem of choosing a metric by which to optimize sample weights, as $\Phi$ is trained to extract information predictive of the outcome. We capture these ideas in an upper bound on the generalization error under a shift in design and specialize it to the case of treatment effect estimation.

**Main contributions**   We bring together two techniques used to overcome distributional shift between designs—re-weighting and representation learning, with complementary robustness properties, generalizing existing methods based on either technique. We give finite-sample generalization bounds for prediction under design shift, *without* assuming access to importance sampling weights *or* to a well-specified model, and develop an algorithmic framework to minimize these bounds. We propose a neural network architecture that jointly learns a representation of the input and a weighting function to improve balance across changing settings. Finally, we apply our proposed algorithm to the task of predicting causal effects from observational data, achieving state-of-the art results on a widely used benchmark.

## 2   PREDICTING OUTCOMES UNDER DESIGN SHIFT

The goal of this work is to accurately predict outcomes of interventions $T \in \mathcal{T}$ in contexts $X \in \mathcal{X}$ drawn from a *target design* $p_\pi(X, T)$. The outcome of intervening with $t \in \mathcal{T}$ is the potential outcome $Y(t) \in \mathcal{Y}$ (Imbens & Rubin, 2015, Ch. 1–2), which has a stationary distribution $p_t(Y \mid X)$ given context $X$. Assuming a stationary outcome is akin to the *covariate shift* assumption (Shimodaira, 2000), often used in domain adaptation.[1] For example, in the classical binary setting, $Y(1)$ represents the outcome under treatment and $Y(0)$ the outcome under control. The target design consists of two components: the *target policy* $p_\pi(T \mid X)$, which describes how one intends to map observations of contexts (such as patient prognostics) to interventions (such as pharmacological treatments) and the *target domain* $p_\pi(X)$, which describes the population of contexts to which the policy will be applied. The target design is known to us only through $m$ unlabeled sam-

---

[1]Equivalently, we may write $p_\pi(Y(t) \mid X) = p_\mu(Y(t) \mid X)$.

ples $(x'_1, t'_1), \ldots, (x'_m, t'_m)$ from $p_\pi(X, T)$. Outcomes are only available to us in labeled samples from a source domain: $(x_1, t_1, y_1), \ldots, (x_n, t_n, y_n)$, where $(x_i, t_i)$ are draws from a source design $p_\mu(X, T)$ and $y_i = y_i(t_i)$ is a draw from $p_T(Y \mid X)$, corresponding only to the *factual* outcome $Y(T)$ of the treatment administered. Like the target design, the source design consists of a domain of contexts for which we have data and a policy, which describes the (unknown) historical administration of treatment in the data. Only the factual outcomes of the treatments administered are observed, while the counterfactual outcomes $y_i(t)$ for $t \neq t_i$ are, naturally, unobserved.

Our focus is the *observational* or *off-policy* setting, in which interventions in the source design are performed non-randomly as a function of $X$, $p_\mu(T \mid X) \neq p_\mu(T)$. This encapsulates both the covariate shift often observed between treated and control populations in observational studies and the covariate shift between the domain of the study and the domain of an eventual wider intervention. Examples of this problem are plentiful: in addition to the example given in the introduction, consider predicting the return of an advertising policy based on the historical results of a different policy, applied to a different population of customers. We stress that we are interested in the *causal effect* of an intervention $T$ on $Y$, conditioned on $X$. As such, we cannot think of $X$ and $T$ as a single variable. Without additional assumptions, it is impossible to deduce the effect of an intervention based on observational data alone (Pearl, 2009), as it amounts disentangling correlation and causation. Crucially, for any unit $i$, we can observe the potential outcome $y_i(t)$ of at most one intervention $t$. In our analysis, we make the following standard assumptions.

**Assumption 1** (Consistency, ignorability and overlap). *For any unit $i$, assigned to intervention $t_i$, we observe $Y_i = Y(t_i)$. Further, $\{Y(t)\}_{t \in \mathcal{T}}$ and the data-generating process $p_\mu(X, T, Y)$ satisfy strong ignorability: $\{Y(t)\}_{t \in \mathcal{T}} \perp\!\!\!\perp T \mid X$ and overlap: $\mathrm{Pr}_{p_\pi}(p_\mu(T \mid X) > 0) = 1$.*

Assumption 1 is a sufficient condition for *causal identifiability* (Rosenbaum & Rubin, 1983). Ignorability is also known as the *no hidden confounders* assumption, indicating that all variables that cause both $T$ and $Y$ are assumed to be measured. Under ignorability therefore, any domain shift in $p(X)$ cannot be due to variables that causally influence $T$ and $Y$, other than through $X$. Under Assumption 1, potential outcomes equal conditional expectations: $\mathbb{E}[Y(t) \mid X = x] = \mathbb{E}[Y \mid X = x, T = t]$, and we may predict $Y(t)$ by regression. We further assume common domain support, $\forall x \in \mathcal{X} : p_\pi(X = x) > 0 \Rightarrow p_\mu(X = x) > 0$. Finally, we adopt the notation $p(x) := p(X = x)$.

## 2.1 RE-WEIGHTED RISK MINIMIZATION

We attempt to learn predictors $f : \mathcal{X} \times \mathcal{T} \to \mathcal{Y}$ such that $f(x, t)$ approximates $\mathbb{E}[Y \mid X = x, T = t]$. Recall that under Assumption 1, this conditional expectation is equal to the (possibly counterfactual) potential outcome $Y(t)$, conditioned on $X$. Our goal is to ensure that hypotheses $f$ are accurate under a design $p_\pi$ that deviates from the data-generating process, $p_\mu$. This is unlike standard supervised learning for which $p_\pi = p_\mu$. We measure the (in)ability of $f$ to predict outcomes under $\pi$, using the expected risk,

$$R_\pi(f) := \mathbb{E}_{x,t,y \sim p_\pi}[\ell_f(x, t, y)] \tag{1}$$

based on a sample from $\mu$, $D^n_\mu = \{(x_i, t_i, y_i) \sim p_\mu; i = 1, ..., n\}$. Here, $\ell_f(x, t, y) := L(f(x, t), y)$ is an appropriate loss function, such as the squared loss, $L(y, y') := (y - y')^2$ or the log-loss, depending on application. As outcomes under the target design $p_\pi$ are not observed, even through a finite sample, we cannot directly estimate (1) using the empirical risk under $p_\pi$. A common way to resolve this is to use *importance sampling* (Shimodaira, 2000)—the observation that if $p_\mu$ and $p_\pi$ have common support, with $w^*(x, t) = p_\pi(x, t)/p_\mu(x, t)$,

$$R^{w^*}_\mu(f) := \mathbb{E}_{x,t,y \sim p_\mu}[w^*(x, t)\ell_f(x, t, y)] = R_\pi(f) . \tag{2}$$

Hence, with access to $w^*$, an unbiased estimator of $R_\pi(f)$ may be obtained by re-weighting the (factual) empirical risk under $\mu$,

$$\hat{R}^{w^*}_\mu(f) := \frac{1}{n} \sum_{i=1}^{n} w^*(x_i, t_i)\ell_f(x_i, t_i, y_i) . \tag{3}$$

Unfortunately, importance sampling weights can be very large when $p_\pi$ is large and $p_\mu$ small, resulting in large variance in $\hat{R}^{w^*}_\mu(f)$ (Swaminathan & Joachims, 2015). More importantly, $p_\mu(x, t)$ is rarely known in practice, and neither is $w^*$. In principle, however, any re-weighting function $w$ with the following property yields a valid risk under the re-weighted distribution $p^w_\mu$.

**Definition 1.** *A function $w : \mathcal{X} \times \mathcal{T} \to \mathbb{R}_+$ is a valid re-weighting of $p_\mu$ if*
$$\mathbb{E}_{x,t \sim p_\mu}[w(x,t)] = 1 \ \ \text{and} \ \ p_\mu(x,t) > 0 \Rightarrow w(x,t) > 0.$$
*We denote the re-weighted density $p_\mu^w(x,t) := w(x,t)p_\mu(x,t)$.*

A natural candidate in place of $w^*$ is an estimate $\hat{w}^*$ based on estimating densities $p_\pi(x,t)$ and $p_\mu(x,t)$. In this work, we adopt a different strategy, learning parameteric re-weighting functions $w$ from observational data, that minimize an upper bound on the risk under $p_\pi$.

## 2.2 CONDITIONAL TREATMENT EFFECT ESTIMATION

An important special case of our setting is when treatments are binary, $T \in \{0, 1\}$, often interpreted as treating ($T = 1$) or not treating ($T = 0$) a unit, and the domain is fixed across designs, $p_\mu(X) = p_\pi(X)$. This is the classical setting for estimating treatment effects—the effect of choosing one intervention over another (Morgan & Winship, 2014).[2] The effect of an intervention $T = 1$ in context $X$, is measured by the *conditional average treatment effect* (CATE), $\tau(x) = \mathbb{E}[Y(1) - Y(0) \mid X = x]$. Predicting $\tau$ for unobserved units typically involves prediction of both potential outcomes[3]. In a clinical setting, knowledge of $\tau$ is necessary to assess which medication should be administered to a certain individual. Historically, the (population) *average treatment effect*, ATE $= \mathbb{E}_{x \sim p}[\tau(x)]$, has received comparatively much more attention (Rosenbaum & Rubin, 1983), but is inadequate for personalized decision making. Using predictors $f(x,t)$ of potential outcomes $Y(t)$ in contexts $X = x$, we can estimate the CATE by $\hat{\tau}(x) = f(x,1) - f(x,0)$ and measure the quality using the mean squared error (MSE),
$$\text{MSE}(\hat{\tau}) = \mathbb{E}_p\left[(\hat{\tau}(x) - \tau(x))^2\right] \tag{4}$$
In Section 4, we argue that estimating CATE from observational data requires overcoming distributional shift with respect to the treat-all and treat-none policies, in predicting each respective potential outcome, and show how this can be used to derive generalization bounds for CATE.

## 3 RELATED WORK

A large body of work has shown that under assumptions of ignorability and having a well-specified model, various regression methods for counterfactual estimation are asymptotically consistent (Chernozhukov et al., 2017; Athey & Imbens, 2016; Belloni et al., 2014). However, consistency results like these provide little insight into the case of model misspecification. Under model misspecification, regression methods may suffer from additional bias when generalizing across designs due to distributional shift. A common way to alleviate this is *importance sampling*, see Section 2. This idea is used in propensity-score methods (Austin, 2011), that use treatment assignment probabilities (propensities) to re-weight samples for causal effect estimation, and more generally in re-weighted regression, see e.g. (Swaminathan & Joachims, 2015). A major drawback of these methods is the assumption that the design density is known. To address this, others (Gretton et al., 2009; Kallus, 2016), have proposed *learning* sample weights $w$ to minimize the distributional distance between samples under $p_\pi$ and $p_\mu^w$, but rely on specifying the data representation a priori, without regard for which aspects of the data actually matter for outcome prediction and policy estimation.

On the other hand, Johansson et al. (2016); Shalit et al. (2017) proposed *learning representations* for counterfactual inference, inspired by work in unsupervised domain adaptation (Mansour et al., 2009). The drawback of this line of work is that the generalization bounds of Shalit et al. (2017) and Long et al. (2015) are loose—even if infinite samples are available, they are not guaranteed to converge to the lowest possible error. Moreover, these approaches do not make use of important information that can be estimated from data: the treatment/domain assignment probabilities.

## 4 GENERALIZATION UNDER DESIGN SHIFT

We give a bound on the risk in predicting outcomes under a target design $p_\pi(T, X)$ based on unlabeled samples from $p_\pi$ and labeled samples from a source design $p_\mu(T, X)$. Our result combines representation learning, distribution matching and re-weighting, resulting in a tighter bound

---

[2]Notions of causal effects exist also for the non-binary case, but these are not considered here.
[3]This is sufficient but not necessary.

than the closest related work, Shalit et al. (2017). The predictors we consider are compositions $f(x,t) = h(\Phi(x),t)$ where $\Phi$ is a representation of $x$ and $h$ an hypothesis. We first give an upper bound on the risk in the general design shift setting, then show how this result can be used to bound the error in prediction of treatment effects. In Section 5 we give a result about the asymptotic properties of the minimizers of this upper bound.

**Risk under distributional shift**    Our bounds on the risk under a target design capture the intuition that if either a) the target design $\pi$ and source design $\mu$ are close, or b) the true outcome is a simple function of $x$ and $t$, the gap between the target risk and the re-weighted source risk is small. These notions can be formalized using integral probability metrics (IPM) (Sriperumbudur et al., 2009) that measure distance between distributions w.r.t. a normed vector space of functions $\mathcal{H}$.

**Definition 2.** *The integral probability metric (IPM) distance, associated with a normed vector space of functions $\mathcal{H}$, between distributions $p$ and $q$ is,* $\mathrm{IPM}_{\mathcal{H}}(p,q) := \sup_{h \in \mathcal{H}} |\mathbb{E}_p[h] - \mathbb{E}_q[h]|$.

Important examples of IPMs include the Wasserstein distance, for which $\mathcal{H}$ is the family of functions with Lipschitz constant at most 1, and the Maximum Mean Discrepancy for which $\mathcal{H}$ are functions in the norm-1 ball in a reproducing kernel Hilbert space. Using definitions 1–2, and the definition of re-weighted risk, see (2), we can state the following result (see Appendix A.2 for a proof).

**Lemma 1.** *For hypotheses $f$ with loss $\ell_f$ such that $\ell_f / \|\ell_f\|_{\mathcal{H}} \in \mathcal{H}$, and $p_\mu, p_\pi$ with common support, there exists a valid re-weighting $w$ of $p_\mu$, see Definition 1, such that,*

$$R_\pi(f) \le R_\mu^w(f) + \|\ell_f\|_{\mathcal{H}} \mathrm{IPM}_{\mathcal{H}}(p_\pi, p_\mu^w) \le R_\mu(f) + \|\ell_f\|_{\mathcal{H}} \mathrm{IPM}_{\mathcal{H}}(p_\pi, p_\mu) \,. \qquad (5)$$

*The first inequality is tight for importance sampling weights, $w(x,t) = p_\pi(x,t)/p_\mu(x,t)$. The second inequality is not tight for general $f$, even if $\ell_f / \|\ell_f\|_{\mathcal{H}} \in \mathcal{H}$, unless $p_\pi = p_\mu$.*

The bound of Lemma 1 is tighter if $p_\mu$ and $p_\pi$ are close (the IPM is smaller), and if the loss lives in a small family of functions $\mathcal{H}$ (the supremum is taken over a smaller set). Lemma 1 also implies that there exist weighting functions $w(x,t)$ that achieve a tighter bound than the uniform weighting $w(x,t) = 1$, implicitly used by Shalit et al. (2017). While importance sampling weights result in a tight bound in expectation, neither the design densities nor their ratio are known in general. Moreover, exact importance weights often result in large variance in finite samples (Cortes et al., 2010). Here, we will search for a weighting function $w$, that minimizes a finite-sample version of (5), trading off bias and variance. We examine the empirical value of this idea alone in Section 6.1. We now introduce the notion of *representation learning* to combat distributional shift.

**Representation learning**    The idea of learning representations that reduce distributional shift in the induced space, and thus the source-target error gap, has been applied in domain adaptation (Ajakan et al., 2014), algorithmic fairness (Zemel et al., 2013) and counterfactual prediction (Shalit et al., 2017). The hope of these approaches is to learn predictors that predominantly exploit information that is common to both source and target distributions. For example, a face detector should be able to recognize the structure of human features even under highly variable environment conditions, by ignoring background, lighting etc. We argue that re-weighting (e.g. importance sampling) should also only be done with respect to features that are predictive of the outcome. Hence, in Section 5, we propose using re-weightings that are functions of learned representations.

We follow the setup of Shalit et al. (2017), and consider learning twice-differentiable, invertible representations $\Phi : \mathcal{X} \to \mathcal{Z}$, where $\mathcal{Z}$ is the representation space, and $\Psi : \mathcal{Z} \to \mathcal{X}$ is the *inverse representation*, such that $\Psi(\Phi(x)) = x$ for all $x$. Let $\mathcal{E}$ denote space of such representation functions. For a design $\pi$, we let $p_{\pi,\Phi}(z,t)$ be the distribution induced by $\Phi$ over $\mathcal{Z} \times \mathcal{T}$, with $p_{\pi,\Phi}^w(z,t) := p_{\pi,\Phi}(z,t)w(\Psi(z),t)$ its re-weighted form and $\hat{p}_{\pi,\Phi}^w$ its re-weighted empirical form, following our previous notation. Finally, we let $\mathcal{G} \subseteq \{h : \mathcal{Z} \times \mathcal{T} \to \mathcal{Y}\}$ denote a set of hypotheses $h(\Phi,t)$ operating on the representation $\Phi$ and let $\mathcal{F}$ the space of all compositions, $\mathcal{F} = \{f = h(\Phi(x),t) : h \in \mathcal{G}, \Phi \in \mathcal{E}\}$. We can now relate the expected target risk $R_\pi(f)$ to the re-weighted empirical source risk $\hat{R}_\mu^w(f)$.

**Theorem 1.** *Given is a labeled sample $(x_1,t_1,y_1),...,(x_n,t_n,y_n)$ from $p_\mu$, and an unlabeled sample $(x'_1,t'_1),...,(x'_m,t'_m)$ from $p_\pi$, with corresponding empirical measures $\hat{p}_\mu$ and $\hat{p}_\pi$. Suppose that $\Phi$ is a twice-differentiable, invertible representation, that $h(\Phi,t)$ is an hypothesis, and $f = h(\Phi(x),t) \in \mathcal{F}$. Define $m_t(x) = \mathbb{E}_Y[Y \mid X = x, T = t]$, let $\ell_{h,\Phi}(\Psi(z),t) :=$*

$L(h(z,t), m_t(\Psi(z)))$ where $L$ is the squared loss, $L(y, y') = (y - y')^2$, and assume that there exists a constant $B_\Phi > 0$ such that $\ell_{h,\Phi}/B_\Phi \in \mathcal{H} \subseteq \{h : \mathcal{Z} \times \mathcal{T} \to \mathcal{Y}\}$, where $\mathcal{H}$ is a reproducing kernel Hilbert space of a kernel, $k$ such that $k((z,t),(z,t)) < \infty$. Finally, let $w$ be a valid re-weighting of $p_{\mu,\Phi}$. Then with probability at least $1 - 2\delta$,

$$R_\pi(f) \leq \hat{R}_\mu^w(f) + B_\Phi \mathrm{IPM}_\mathcal{H}(\hat{p}_{\pi,\Phi}, \hat{p}_{\mu,\Phi}^w) + V_\mu(w, \ell_f)\frac{\mathcal{C}_{n,\delta}^\mathcal{F}}{n^{3/8}} + \mathcal{D}_\delta^{\Phi,\mathcal{H}}\left(\frac{1}{\sqrt{m}} + \frac{1}{\sqrt{n}}\right) + \sigma_Y^2 \quad (6)$$

where $\mathcal{C}_{n,\delta}^\mathcal{F}$ is a function of the pseudo-dimension of $\mathcal{F}$, $\mathcal{D}_{m,n,\delta}^\mathcal{H}$ a function of the kernel norm of $\mathcal{H}$, both only with logarithmic dependence on $n$ and $m$, $\sigma_Y^2$ is the expected variance in $Y$, and

$$V_\mu(w, \ell_f) = \max\left(\sqrt{\mathbb{E}_{p_\mu}[w^2(x,t)\ell_f^2(x,t)]}, \sqrt{\mathbb{E}_{\hat{p}_\mu}[w^2(x,t)\ell_f^2(x,t)]}\right).$$

A similar bound exists where $\mathcal{H}$ is the family of functions Lipschitz constant at most 1, and $\mathrm{IPM}_\mathcal{H}$ the Wasserstein distance, but with worse sample complexity.

See Appendix A.2 for a proof of Theorem 1 that involves applying finite-sample generalization bounds to Lemma 1, as well as moving to the space induced by the representation $\Phi$.

Theorem 1 has several implications: non-identity feature representations, non-uniform sample weights, and variance control of these weights can all contribute to a lower bound. Using uniform weights $w(x,t) = 1$ in (6), results in a bound similar to that of Shalit et al. (2017) and Long et al. (2015). When $\pi \neq \mu$, minimizing uniform-weight bounds results in biased hypotheses, even in the asymptotical limit, as the IPM term does not vanish when the sample size increases. This is an undesirable property, as even $k$-nearest-neighbor classifiers are consistent in the limit of infinite samples. We consider minimizing (6) with respect to $w$, improving the tightness of the bound.

Theorem 1 indicates that even though importance sampling weights $w^*$ yield estimators with small bias, they can suffer from high variance, as captured by the factor $V_\mu(w, \ell_f)$. The factor $B_\Phi$ is not known in general as it depends on the true outcome, and is determined by $\|\ell_f\|_\mathcal{H}$ as well as the determinant of the Jacobian of $\Psi$, see Appendix A.2. Qualitatively, $B_\Phi$ measures the joint complexity of $\Phi$ and $\ell_f$ and is sensitive to the scale of $\Phi$—as the scale of $\Phi$ vanishes, $B_\Phi$ blows up. To prevent this in practice, we normalize $\Phi$. As $B_\Phi$ is unknown, Shalit et al. (2017) substituted a hyperparameter $\alpha$ for $B_\Phi$, but discussed the difficulties of selecting its value without access to counterfactual labels. In our experiments, we explore a heuristic for adaptively choosing $\alpha$, based on measures of complexity of the observed held-out loss as a function of the input. Finally, the term $\mathcal{C}_{n,\delta}^\mathcal{F}$ follows from standard learning theory results (Cortes et al., 2010) and $\mathcal{F}$, and $\mathcal{D}_\delta^{\Phi,\mathcal{H}}$ from concentration results for estimating IPMs (Sriperumbudur et al., 2012), see Appendix A.2.

Theorem 1 is immediately applicable to the case of unsupervised domain adaptation in which there is only a single potential outcome of interest, $\mathcal{T} = \{0\}$. In this case, $p_\mu(T \mid X) = p_\pi(T \mid X) = 1$.

**Conditional average treatment effects** A simple argument shows that the error in predicting the conditional average treatment effect, $\mathrm{MSE}(\hat{\tau})$ can be bounded by the sum of risks under the constant treat-all and treat-none policies. As in Section 2.2, we consider the case of a fixed domain $p_\pi(X) = p_\mu(X)$ and binary treatment $\mathcal{T} = \{0,1\}$. Let $R_{\pi_t}(f)$ denote the risk under the constant policy $\pi_t$ such that $\forall x \in \mathcal{X} : p_{\pi_t}(T = t \mid X = x) = 1$.

**Proposition 1.** We have with $\mathrm{MSE}(\hat{\tau})$ as in (4) and $R_{\pi_t}(f)$ the risk under the constant policy $\pi_t$,

$$\mathrm{MSE}(\hat{\tau}) \leq 2(R_{\pi_1}(f) + R_{\pi_0}(f)) - 4\sigma^2 \quad (7)$$

where $\sigma$ is such that $\forall t \in \mathcal{T}, x \in \mathcal{X}, \sigma_Y(x,t) \geq \sigma$ and $\sigma_Y(x,t)$ is standard deviation of $Y(t)$ conditioned on $X = x$.

The proof involves the relaxed triangle inequality and the law of total probability. By Proposition 1, we can apply Theorem 1 to $R_{\pi_1}$ and $R_{\pi_0}$ separately, to obtain a bound on $\mathrm{MSE}(\tau)$. For brevity, we refrain from stating the full result, but emphasize that it follows from Theorem 1. In Section 6.2, we evaluate our framework in treatment effect estimation, minimizing this bound.

## 5 JOINT LEARNING OF REPRESENTATIONS AND SAMPLE WEIGHTS

Motivated by the theoretical insights of Section 4, we propose to jointly learn a representation $\Phi(x)$, a re-weighting $w(x,t)$ and an hypothesis $h(\Phi, t)$ by minimizing a bound on the risk under the target

design, see (6). This approach improves on previous work in two ways: it alleviates the bias of Shalit et al. (2017) when sample sizes are large, see Section 4, and it increases the flexibility of the balancing method of Gretton et al. (2009) by learning the representation to balance.

For notational brevity, we let $w_i = w(x_i, t_i)$. Recall that $\hat{p}_{\pi,\Phi}^w$ is the re-weighted empirical distribution of representations $\Phi$ under $p_\pi$. The training objective of our algorithm is the RHS of (6), with hyperparameters $\beta = (\alpha, \lambda_h, \lambda_w)$ substituted for model (and representation) complexity terms,

$$\mathcal{L}_\pi(h, \Phi, w; \beta) = \underbrace{\frac{1}{n}\sum_{i=1}^n w_i \ell_h(\Phi(x_i), t_i) + \frac{\lambda_h}{\sqrt{n}}\mathcal{R}(h)}_{\mathcal{L}_\pi^h(h,\Phi,w;D,\alpha,\lambda_h)} + \underbrace{\alpha\, \text{IPM}_G(\hat{p}_{\pi,\Phi}, \hat{p}_{\mu,\Phi}^w) + \lambda_w \frac{\|w\|_2}{n}}_{\mathcal{L}_\pi^w(\Phi,w;D,\alpha,\lambda_w)} \qquad (8)$$

where $\mathcal{R}(h)$ is a regularizer of $h$, such as $\ell_2$-regularization. We can show the following result.

**Theorem 2.** *Suppose $\mathcal{H}$ is a reproducing kernel Hilbert space given by a bounded kernel. Suppose weak overlap holds in that $\mathbb{E}[(p_\pi(x,t)/p_\mu(x,t))^2] < \infty$. Then,*

$$\min_{h,\Phi,w} \mathcal{L}_\pi(h, \Phi, w; \beta) \leq \min_{f \in \mathcal{F}} R_\pi(f) + O_p(1/\sqrt{n} + 1/\sqrt{m}) \ .$$

*Consequently, under the assumptions of Thm. 1, for sufficiently large $\alpha$ and $\lambda_w$,*

$$R_\pi(\hat{f}_n) \leq \min_{f \in \mathcal{F}} R_\pi(f) + O_p(1/n^{3/8} + 1/\sqrt{m}).$$

*In words, the minimizers of* (8) *converge to the representation and hypothesis that minimize the counterfactual risk, in the limit of infinite samples.*

**Implementation** Minimization of $\mathcal{L}_\pi(h, \Phi, w; \beta)$ over $h$, $\Phi$ and $w$ is, while motivated by Theorem 2, a difficult optimization problem to solve in practice. For example, adjusting $w$ to minimize the empirical risk term may result in overemphasizing "easy" training examples, resulting in a poor local minimum. Perhaps more importantly, ensuring invertibility of $\Phi$ is challenging for many representation learning frameworks, such as deep neural networks. In our implementation, we deviate from theory on these points, by fitting the re-weighting $w$ based only on imbalance and variance terms, and don't explicitly enforce invertibility. As a heuristic, we split the objective, see (8), in two and use only the IPM term and regularizer to learn $w$. In short, we adopt the following alternating procedure.

$$h^k, \Phi^k = \underset{h,\Phi}{\arg\min}\ \mathcal{L}_\pi^h(h, \Phi, w; D, \alpha, \lambda_h), \quad w^{k+1} = \underset{w}{\arg\min}\ \mathcal{L}_\pi^w(\Phi^k, w; D, \alpha, \lambda_w) \qquad (9)$$

The re-weighting function $w(x, t)$ could be represented by one free parameter per training point, as it is only used to learn the model, not for prediction. However, we propose to let $w$ be a parametric function of $\Phi(x)$. Doing so ensures that information predictive of the outcome is used for balancing, and lets us compute weights on a hold-out set, to perform early stopping or select hyperparameters. This is not possible with existing re-weighting methods such as Gretton et al. (2009); Kallus (2016). An example architecture for the treatment effect estimation setting is presented in Figure 1. By Proposition 1, estimating treatment effects involves predicting under the two constant policies— treat-everyone and treat-no-one. In Section 6, we evaluate our method in this task.

As noted by Shalit et al. (2017), choosing hyperparameters for counterfactual prediction is fundamentally difficult, as we cannot observe ground truth for counterfactuals. In this work, we explore setting the balance parameter $\alpha$ adaptively. $\alpha$ is used in (8) in place of $B_\Phi$, a factor measuring the complexity of the loss and representation function as functions of the input, a quantity that changes during training. As a heuristic, we use an approximation of the Lipschitz constant of $\ell_f$, with $f = h(\Phi(x), t)$, based on observed examples: $\alpha_{h,\Phi} = \max_{i,j \in [n]} |\ell_f(x_i, t_i, y_i) - \ell_f(x_j, t_j, y_j)|/\|x_i - x_j\|_2$. We use a moving average over batches to improve stability.

# 6 EXPERIMENTS

## 6.1 SYNTHETIC EXPERIMENTS FOR DOMAIN ADAPTATION

We create a synthetic domain adaptation experiment to highlight the benefit of using a learned re-weighting function to minimize weighted risk over using importance sampling weights $w^*(x) =$

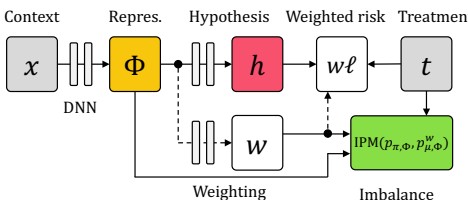

Context  Repres.  Hypothesis  Weighted risk  Treatment

DNN

Weighting        Imbalance

Figure 1: Architecture for predicting outcomes under design shift. A re-weighting function $w$ is fit jointly with a representation $\Phi$ and hypothesis $h$ of the potential outcomes, to minimize a bound on the target risk. Dashed lines are not back-propagated through. Regularization penalties not shown.

Table 1: Causal effect estimation on IHDP. CATE error RMSE$(\hat{\tau})$, target prediction error $\hat{R}_\pi(f)$ and std errors. Lower is better.

|  | RMSE$(\hat{\tau})$ | $\hat{R}_\pi(f)$ |
|---|---|---|
| OLS | $2.3 \pm .11$ | $1.1 \pm .05$ |
| OLS-IPW | $2.4 \pm .11$ | $1.2 \pm .05$ |
| Random For. | $6.6 \pm .30$ | $4.1 \pm .18$ |
| Causal For. | $3.8 \pm .18$ | $1.8 \pm .08$ |
| BART | $2.3 \pm .10$ | $1.7 \pm .07$ |
| IPM-WNN | $1.2 \pm .12$ | $.65 \pm .02$ |
| CFRW | $.76 \pm .02$ | $.46 \pm .01$ |
| RCFR Oracle $\alpha, w = 1$ | $.81 \pm .07$ | $.47 \pm .03$ |
| RCFR Oracle $\alpha$ | $.65 \pm .04$ | $.38 \pm .01$ |
| RCFR Adapt. $\alpha$ | $.67 \pm .05$ | $.37 \pm .01$ |

$p_\pi(x)/p_\mu(x)$ for small sample sizes. We observe $n$ labeled source samples, distributed according to $p_\mu(x) = \mathcal{N}(x; m_\mu, I_d)$ and predict for $n$ unlabeled target samples drawn according to $p_\pi(x) = \mathcal{N}(x; m_\pi, I_d)$ where $I_d$ is the $d$-dimensional identity matrix, $m_\mu = b\mathbf{1}_d/2$, $m_\pi = -b\mathbf{1}_d/2$ and $\mathbf{1}_d$ is the $d$-dimensional vector of all 1:s, here with $b = 1$, $d = 10$. We let $\beta \sim \mathcal{N}(\mathbf{0}_d, 1.5I_d)$ and $c \sim \mathcal{N}(0, 1)$ and let $y = \sigma(\beta^\top x + c)$ where $\sigma(z) = 1/(1 + e^{-z})$. Importance sampling weights, $w^*(x) = p_\pi(x)/p_\mu(x)$, are known. In experiments, we vary $n$ from 10 to 600. We fit (misspecified) linear models[4] $f(x) = \beta^\top x + \gamma$ to the logistic outcome, and compare minimizing a weighted source risk by a) parameterizing sample weights as a small feed-forward neural network to minimize (8) (ours) b) using importance sampling weights (baseline), both using gradient descent. For our method, we add a small variance penalty, $\lambda_w = 10^{-3}$, to the learned weights, use MMD with an RBF-kernel of $\sigma = 1.0$ as IPM, and let $\alpha = 10$. We compare to exact importance sampling weights (IS) as well as clipped IS weights (ISC), $w_M(x) = \min(w(x), M)$ for $M \in \{5, 10\}$, a common way of reducing variance of re-weighting methods (Swaminathan & Joachims, 2015).

In Figure 2a, we see that our proposed method behaves well at small sample sizes compared to importance sampling methods. The poor performance of exact IS weights is expected at smaller samples, as single samples are given very large weight, resulting in hypotheses that are highly sensitive to the training set. While clipped weights alleviates this issue, they do not preserve relevance ordering of high-weight samples, as many are given the truncation value $M$, in contrast to the re-weighting learned by our method. True domain densities are known only to IS methods.

## 6.2 CONDITIONAL AVERAGE TREATMENT EFFECTS — IHDP

We evaluate our framework in the CATE estimation setting, see Section 2.2. Our task is to predict the expected difference between potential outcomes conditioned on pre-treatment variables, *for a held-out sample* of the population. We compare our results to ordinary least squares (OLS) (with one regressor per outcome), OLS-IPW (re-weighted OLS according to a logistic regression estimate of propensities), Random Forests, Causal Forests (Wager & Athey, 2017), BART (Chipman et al., 2010), and CFRW (Shalit et al., 2017) (with Wasserstein penalty). Finally, we use as baseline (IPM-WNN): first weights are found by IPM minimization in the input space (Gretton et al., 2009; Kallus, 2016), then used in a re-weighted neural net regression, with the same architecture as our method.

Our implementation, dubbed RCFR for *Re-weighted CounterFactual Regression*, parameterizes representations $\Phi(x)$, weighting functions $w(\Phi, t)$ and hypotheses $h(\Phi, t)$ using neural networks, trained by minimizing (8). We use the RBF-kernel maximum mean discrepancy as the IPM (Gretton et al., 2012). For a description of the architecture, training procedure and hyperparameters, see Appendix B. We compare results using uniform $w = 1$ and learned weights, setting the balance parameter $\alpha$ either fixed, by an oracle (test-set error), or adaptively using the heuristic described in Section 5. To pick other hyperparameters, we split training sets into one part used for function fitting and one used for early stopping and hyperparameter selection. Hyperparameters for regularization are chosen based on the empirical loss on a held-out source (factual) sample.

---

[4]The identity representation $\Phi(x) = x$ is used for both approaches.

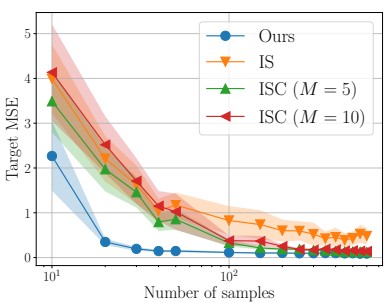 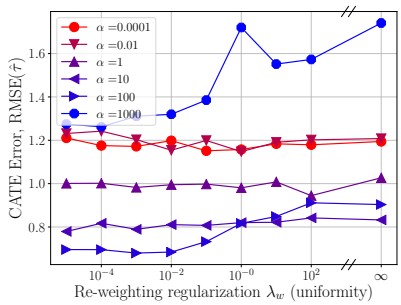

(a) Target prediction error on synthetic domain adaptation experiment, comparing learned re-weighting (RCFR) and exact/clipped importance sampling weights (IS/ISC). The high variance of IS hurts performance for small sample sizes.

(b) For small imbalance penalties $\alpha$, re-weighting (low $\lambda_w$) has no effect. For moderate $\alpha$, less uniform re-weighting (smaller $\lambda_w$) improves the error, c) for large $\alpha$, weighting helps, but overall error increases. Best viewed in color.

Figure 2: Results for domain adaptation and causal effect estimation experiments.

The Infant Health and Development Program (IHDP) dataset is a semi-synthetic binary-treatment benchmark (Hill, 2011), split into training and test sets by Shalit et al. (2017). IHDP has a set of $d = 25$ real-world continuous and binary features describing $n = 747$ children and their mothers, a real-world binary treatment made non-randomized through biased subsampling by Hill (2011), and a synthesized continuous outcome that can be used to compute the ground-truth CATE error. Average results over 100 different realizations/settings of the outcome are presented in Table 1. We see that our proposed method achieves state-of-the-art results, and that adaptively choosing $\alpha$ does not hurt performance much. Furthermore, we see a substantial improvement from using non-uniform sample weights. In Figure 2b we take a closer look at the behavior of our model as we vary its hyperparameters on the IHDP dataset. Between the two plots we can draw the following conclusions: a) For moderate to large $\alpha \in [10, 100]$, we observe a marginal gain from using the IPM penalty. This is consistent with the observations of Shalit et al. (2017). b) For large $\alpha \in [10, 1000]$, we see a large gain from using a non-uniform re-weighting (small $\lambda_w$). c) While large $\alpha$ makes the factual error more representative of the counterfactual error, using it without re-weighting results in higher absolute error. We believe that the moderate sample size of this dataset is one of the reasons for the usefulness of our method. See Appendix C.2 for a complementary view of these results.

## 7 DISCUSSION

We have proposed a theory and an algorithmic framework for learning to predict outcomes of interventions under shifts in design—changes in both intervention policy and feature domain. The framework combines representation learning and sample re-weighting to balance source and target designs, emphasizing information from the source sample relevant for the target. Existing re-weighting methods either use pre-defined weights or learn weights based on a measure of distributional distance in the input space. These approaches are highly sensitive to the choice of metric used to measure balance, as the input may be high-dimensional and contain information that is not predictive of the outcome. In contrast, by learning weights to achieve balance in representation space, we base our re-weighting only on information that is predictive of the outcome. In this work, we apply this framework to causal effect estimation, but emphasize that joint representation learning and re-weighting is a general idea that could be applied in many applications with design shift.

Our work suggests that distributional shift should be measured and adjusted for in a representation space relevant to the task at hand. Joint learning of this space and the associated re-weighting is attractive, but several challenges remain, including optimization of the full objective and relaxing the invertibility constraint on representations. For example, variable selection methods are not covered by our current theory, as they induce a non-ivertible representation, but a similar intuition holds there—only predictive attributes should be used when measuring imbalance. We believe that addressing these limitations is a fruitful path forward for future work.

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

APPENDIX

# A    PROOFS

## A.1    DEFINITIONS

**Distribution re-weighting**

**Definition 1 (Restated).** *A function $w : \mathcal{X} \times \mathcal{T} \to \mathbb{R}_+$ is a valid re-weighting of $p_\mu$ if*

$$\mathbb{E}_{x,t \sim p_\mu}[w(x,t)] = 1 \quad and \quad p_\mu(x,t) > 0 \Rightarrow w(x,t) > 0.$$

*We denote the re-weighted density $p_\mu^w(x,t) := w(x,t)p_\mu(x,t)$.*

**Expected & empirical risk**    We let the (expected) risk of $f$ measured by $\ell_h$ under $p_\mu$ be denoted

$$R_\mu(h) = \mathbb{E}_{p_\mu}[l_h(x,t)]$$

where $l_h$ is an appropriate loss function, and the empirical risk over a sample $D_\mu = \{(x_1, t_1, y_1)..., (x_n, t_n, y_n)\}$ from $p_\mu$

$$\hat{R}_\mu(f) = \frac{1}{n} \sum_{i=1}^{n} l_f(x_i, t_i, y_i) \,.$$

We use the superscript $w$ to denote the re-weighted risks

$$R_\mu^w(f) = \mathbb{E}[w(x,t)l_f(x,t)] \quad \hat{R}_\mu^w(f) = \frac{1}{n} \sum_{i=1}^{n} w(x_i, t_i)l_h(x_i, t_i, y_i)$$

**Definition A1** (Importance sampling). *For two distributions $p, q$ on $\mathcal{Z}$, of common support, $\forall z \in \mathcal{Z} : p(z) > 0 \iff q(z) > 0$, we call*

$$w_{IS}(z) := \frac{q(z)}{p(z)}$$

*the* importance sampling *weights of $p$ and $q$.*

**Definition 2 (Restated).** *The integral probability metric (IPM) distance, associated with the function family $\mathcal{H}$, between distributions $p$ and $q$ is defined by*

$$\text{IPM}_\mathcal{H}(p, q) := \sup_{h:\|h\|_\mathcal{H}=1} |\mathbb{E}_p[h] - \mathbb{E}_q[h]|$$

## A.2    LEARNING BOUNDS

We begin by bounding the expected risk under a distribution $p_\pi$ in terms of the expected risk under $p_\mu$ and a measure of the discrepancy between $p_\pi$ and $p_\mu$. Using definition 2 we can show the following result.

**Lemma 1 (Restated).** *For hypotheses $f$ with loss $\ell_f$ such that $\ell_f/\|\ell_f\|_\mathcal{H} \in \mathcal{H}$, and $p_\mu, p_\pi$ with common support, there exists a valid re-weighting $w$ of $p_\mu$, see Definition 1, such that,*

$$R_\pi(f) \leq R_\mu^w(f) + \|\ell_f\|_\mathcal{H}\text{IPM}_\mathcal{H}(p_\pi, p_\mu^w) \leq R_\mu(f) + \|\ell_f\|_\mathcal{H}\text{IPM}_\mathcal{H}(p_\pi, p_\mu) \,. \tag{10}$$

*The first inequality is tight for importance sampling weights, $w(x,t) = p_\pi(x,t)/p_\mu(x,t)$. The second inequality is not tight for general $f$, even if $\ell_f \in \mathcal{H}$, unless $p_\pi = p_\mu$.*

*Proof.* The results follows immediately from the definition of IPM.

$$R_\pi(f) - R_\mu^w(f) = \mathbb{E}_\pi[\ell_f(x,t)] - \mathbb{E}_\mu[w(x,t)\ell_f(x,t)]$$
$$\leq \sup_{h \in \mathcal{H}_\ell} |\mathbb{E}_\pi[h(x,t)] - \mathbb{E}_\mu[w(x,t)h(x,t)]|$$
$$= \text{IPM}_{\mathcal{H}_\ell}(p_\pi, p_\mu^w)$$

Further, for importance sampling weights $w_{IS}(x,t) = \frac{\pi(t;x)}{\mu(t;x)}$, for any $h \in \mathcal{H}$,

$$\mathbb{E}_\pi[h(x,t)] - \mathbb{E}_\mu[w_{IS}(x,t)h(x,t)] = \mathbb{E}_\pi[h(x,t)] - \mathbb{E}_\mu[\frac{\pi(t;x)}{\mu(t;x)}h(x,t)]$$
$$= 0$$

and the LHS is tight. $\qquad\square$

We could apply Lemma 1 to bound the loss under a distribution $q$ based on the weighted loss under $p$. Unfortunately, bounding the expected risk in terms of another expectation is not enough to reason about generalization from an empirical sample. To do that we use Corollary 2 of Cortes et al. (2010), restated as a Theorem below.

**Theorem A1** (Generalization error of re-weighted loss (Cortes et al., 2010)). *For a loss function $\ell_h$ of any hypothesis $h \in \mathcal{H} \subseteq \{h' : \mathcal{X} \to \mathbb{R}\}$, such that $d = \text{Pdim}(\{\ell_h : h \in \mathcal{H}\})$ where $\text{Pdim}$ is the pseudo-dimension, and a weighting function $w(x)$ such that $\mathbb{E}_p[w] = 1$, with probability $1 - \delta$ over a sample $(x_1, ..., x_n)$, with empirical distribution $\hat{p}$,*

$$R_p^w(h) \le \hat{R}_p^w(h) + 2^{5/4} V_{p,\hat{p}}[w(x)l(x)] \left( \frac{d \log \frac{2ne}{d} + \log \frac{4}{\delta}}{n} \right)^{3/8}$$

*with $V_{p,\hat{p}}[w(x)l(x)] = \max(\sqrt{\mathbb{E}_p[w^2(x)\ell_h^2(x)]}, \sqrt{\mathbb{E}_{\hat{p}}[w^2(x)\ell_h^2(x)]})$. With*

$$\mathcal{C}_n^{\mathcal{H}} = 2^{5/4} \left( d \log \frac{2ne}{d} + \log \frac{4}{\delta} \right)^{3/8}$$

*we get the simpler form*

$$R_p^w(h) \le \hat{R}_p^w(h) + V_{p,\hat{p}}[w(x)l(x)] \frac{\mathcal{C}_n^{\mathcal{H}}}{n^{3/8}} .$$

We will also need the following result about estimating IPMs from finite samples from Sriperumbudur et al. (2009).

**Theorem A2** (Estimation of IPMs from empirical samples (Sriperumbudur et al., 2009)). *Let $M$ be a measurable space. Suppose $k$ is measurable kernel such that $\sup_{x \in M} k(x,x) \le C \le \infty$ and $\mathcal{H}$ the reproducing kernel Hilbert space induced by $k$, with $\nu := \sup_{x \in M, f \in \mathcal{H}} f(x) < \infty$. Then, with $\hat{p}, \hat{q}$ the empirical distributions of $p, q$ from $m$ and $n$ samples respectively, and with probability at least $1 - \delta$,*

$$|\text{IPM}_{\mathcal{H}}(p,q) - \text{IPM}_{\mathcal{H}}(\hat{p},\hat{q})| \le \sqrt{18\nu^2 \log \frac{4}{\delta} C} \left( \frac{1}{\sqrt{m}} + \frac{1}{\sqrt{n}} \right)$$

We consider learning twice-differentiable, invertible representations $\Phi : \mathcal{X} \to \mathcal{Z}$, where $\mathcal{Z}$ is the representation space, and $\Psi : \mathcal{Z} \to \mathcal{X}$ is the inverse representation, such that $\Psi(\Phi(x)) = x$ for all $x$. Let $\mathcal{E}$ denote space of such representation functions. For a design $\pi$, we let $p_{\pi,\Phi}(z,t)$ be the distribution induced by $\Phi$ over $\mathcal{Z} \times \mathcal{T}$, with $p_{\pi,\Phi}^w(z,t) := p_{\pi,\Phi}(z,t)w(\Psi(z),t)$ its re-weighted form and $\hat{p}_{\pi,\Phi}^w$ its re-weighted empirical form, following our previous notation. Note that we do not include $t$ in the representation itself, although this could be done in principle. Let $\mathcal{G} \subseteq \{h : \mathcal{Z} \times \mathcal{T} \to \mathcal{Y}\}$ denote a set of hypotheses $h(\Phi, t)$ operating on the representation $\Phi$ and let $\mathcal{F}$ denote the space of all compositions, $\mathcal{F} = \{f = h(\Phi(x), t) : h \in \mathcal{G}, \Phi \in \mathcal{E}\}$. We now restate and prove Theorem 1.

**Theorem 1 (Restated).** *Given is a labeled sample $D_\mu = \{(x_1, t_1, y_1), ..., (x_n, t_n, y_n)\}$ from $p_\mu$, and an unlabeled sample $D_\pi = \{(x'_1, t'_1), ..., (x'_m, t'_m)\}$ from $p_\pi$, with corresponding empirical measures $\hat{p}_\mu$ and $\hat{p}_\pi$. Suppose that $\Phi$ is a twice-differentiable, invertible representation, that $h(\Phi, t)$ is an hypothesis, and $f = h(\Phi(x), t) \in \mathcal{F}$. Define $m_t(x) = \mathbb{E}_Y[Y \mid X = x, T = t]$, let $\ell_{h,\Phi}(\Psi(z), t) := L(h(z,t), m_t(\Psi(z)))$ where $L$ is the squared loss, $L(y, y') = (y - y')^2$, and assume that there exists a constant $B_\Phi > 0$ such that $\ell_{h,\Phi}/B_\Phi \in \mathcal{H} \subseteq \{h : \mathcal{Z} \times \mathcal{T} \to \mathcal{Y}\}$, where*

$\mathcal{H}$ *is a reproducing kernel Hilbert space of a kernel,* $k$ *such that* $k((z,t),(z,t)) < \infty$. *Finally, let* $w$ *be a valid re-weighting of* $p_{\mu,\Phi}$. *Then with probability at least* $1 - 2\delta$,

$$R_\pi(f) \le \hat{R}_\mu^w(f) + B_\Phi \mathrm{IPM}_\mathcal{H}(\hat{p}_{\pi,\Phi}, \hat{p}_{\mu,\Phi}^w) + V_\mu(w, \ell_f) \frac{\mathcal{C}_{n,\delta}^\mathcal{F}}{n^{3/8}} + \mathcal{D}_\delta^{\Phi,\mathcal{H}} \left( \frac{1}{\sqrt{m}} + \frac{1}{\sqrt{n}} \right) + \sigma_Y^2 \quad (11)$$

*where* $\mathcal{C}_{n,\delta}^\mathcal{F}$ *measures the capacity of* $\mathcal{F}$ *and has only logarithmic dependence on* $n$, $\mathcal{D}_{m,n,\delta}^\mathcal{H}$ *measures the capacity of* $\mathcal{H}$, $\sigma_Y^2$ *is the expected variance in potential outcomes, and*

$$V_\mu(w, \ell_f) = \max(\sqrt{\mathbb{E}_{p_\mu}[w^2(x,t)\ell_f^2(x,t)]}, \sqrt{\mathbb{E}_{\hat{p}_\mu}[w^2(x,t)\ell_f^2(x,t)]}) \ .$$

*A similar bound exists where* $\mathcal{H}$ *is the family of functions Lipschitz constant at most 1, but with worse sample complexity.*

*Proof.* We have by definition

$$R_\pi(f) - R_\mu^w(f) = \mathbb{E}_\pi[\ell_f(x,t,y)] - \mathbb{E}_\mu[w(x,t)\ell_f(x,t,y)]$$
$$= \int_{x,t,y} \ell_f(x,t,y) p(y \mid t, x)(p_\pi(x,t) - p_\mu^w(x,t)) dx dt dy$$

Define $\ell_{h,\Phi}(x,t) = L(h(\Phi(x),t), m_t(x))$ where $m_t(x) := E[Y \mid T = t, X = x])$. Then, with $L$, the squared loss, $L(y,y') = (y - y')^2$, we have,

$$\mathbb{E}_\pi[\ell_{h,\Phi}(x,t,y)] = \mathbb{E}_\pi[\ell_{h,\Phi}(x,t)] + \sigma_\pi^2$$

where $\sigma_\pi^2 = \mathbb{E}_{p_\pi}[(Y - m_t(x))^2]$, and analogously for $\mu$. We get that

$$R_\pi(f) - R_\mu^w(f) = \int_{x \in \mathcal{X}, t \in \mathcal{T}} \ell_{h,\Phi}(x,t)(p_\pi(x,t) - p_\mu^w(x,t)) dx dt + \sigma_\pi^2 + \sigma_\mu^2$$
$$= \int_{z \in \mathcal{Z}, t \in \mathcal{T}} \ell_{h,\Phi}(\Psi(z),t)(p_{\pi,\Phi}(z,t) - p_{\mu,\Phi}^w(z,t)) |J_\Psi(z)| dz dt + \sigma_\pi^2 + \sigma_\mu^2$$
$$\le A_\Phi \int_{z \in \mathcal{Z}, t \in \mathcal{T}} \ell_{h,\Phi}(\Psi(z),t)(p_\pi(z,t) - p_\mu^w(z,t)) dz dt + \sigma_\pi^2 + \sigma_\mu^2$$
$$\le \sigma_\pi^2 + \sigma_\mu^2 + A_\Phi \|\ell_{h,\Phi}\|_\mathcal{H} \sup_{h \in \mathcal{H}} \left| \int_{\substack{z \in \mathcal{Z} \\ t \in \mathcal{T}}} h(\Psi(z),t) \left( p_{\pi,\Phi}(rz,t) - p_{\mu,\Phi}^w(z,t) \right) dz dt \right|$$
$$= B_\Phi \cdot \mathrm{IPM}_\mathcal{H}(p_{\pi,\Phi}, p_{\mu,\Phi}^w) + \sigma_\pi^2 + \sigma_\mu^2$$

where $J_\Psi(z)$ is the Jacobian matrix of $\Psi$ evaluated at $z$ and $A_\Phi \ge |J_\Psi(z)|$ for all $z \in \mathcal{Z}$, where $|J|$ is the absolute determinant of $J$. By application of Theorem A1 we have with probability at least $1 - \delta$,

$$R_\mu^w(f) \le \hat{R}_\mu^w(f) + V_\mu(w, \ell) \frac{\mathcal{C}_{n,\delta}^\mathcal{H}}{n^{3/8}} \ .$$

and by applying Theorem A2, we have with probability at least $1 - \delta$,

$$\left| \mathrm{IPM}_\mathcal{H}(p_{\pi,\Phi}, p_{\mu,\Phi}^w) - \mathrm{IPM}_\mathcal{H}(\hat{p}_{\pi,\Phi}, \hat{p}_{\mu,\Phi}^w) \right| \le \sqrt{18\nu^2 \log \frac{4}{\delta} C} \left( \frac{1}{\sqrt{m}} + \frac{1}{\sqrt{n}} \right)$$

We let $\sigma_Y^2 = \sigma_\pi^2 + \sigma_\mu^2$ and

$$\mathcal{D}_\delta^{\Phi,\mathcal{H}} := B_\Phi \sqrt{18\nu^2 \log \frac{4}{\delta} C}$$

Combining these results, observing that $(1 - \delta)^2 \ge 1 - 2\delta$, we obtain the desired result. $\square$

### A.3 Asymptotics

**Theorem 2 (Restated).** *Suppose $\mathcal{H}$ is a reproducing kernel Hilbert space given by a bounded kernel. Suppose weak overlap holds in that $\mathbb{E}[(p_\pi(x,t)/p_\mu(x,t))^2] < \infty$. Then,*

$$\min_{h,\Phi,w} \mathcal{L}_\pi(h,\Phi,w;\beta)] \le \min_{f\in\mathcal{F}} R_\pi(f) + O(1/\sqrt{n} + 1/\sqrt{m}) .$$

*Proof.* Let $f^* = \Phi^* \circ h^* \in \arg\min_{f\in\mathcal{F}} R_\pi(f)$ and let $w^*(x,t) = p_{\pi,\Phi}(\Phi^*(x),t)/p_{\mu,\Phi}(\Phi^*(x),t)$. Since $\min_{h,\Phi,w} \mathcal{L}_\pi(h,\Phi,w;\beta) \le \mathcal{L}_\pi(h^*,\Phi^*,w^*;\beta)$, it suffices to show that $\mathcal{L}_\pi(h^*,\Phi^*,w^*;\beta) = R_\pi(f^*) + O(1/\sqrt{n} + 1/\sqrt{m})$. We will work term by term:

$$\mathcal{L}_\pi(h^*,\Phi^*,w^*;\beta) = \underbrace{\frac{1}{n}\sum_{i=1}^n w_i \ell_h(\Phi(x_i),t_i)}_{\text{\textcircled{A}}} + \underbrace{\lambda_h \frac{\mathcal{R}(h)}{\sqrt{n}}}_{\text{\textcircled{B}}} + \underbrace{\alpha \; \text{IPM}_G(\hat{q}_\Phi, \hat{p}_\Phi^{w^k})}_{\text{\textcircled{C}}} + \underbrace{\lambda_w \frac{\|w\|_2}{n}}_{\text{\textcircled{D}}} .$$

For term $\text{\textcircled{D}}$, letting $w_i^* = w^*(x_i,t_i)$, we have that by weak overlap

$$\text{\textcircled{D}}^2 = \frac{1}{n} \times \frac{1}{n}\sum_{i=1}^n (w_i^*)^2 = O_p(1/n),$$

so that $\text{\textcircled{D}} = O_p(1/\sqrt{n})$. For term $\text{\textcircled{A}}$, under ignorability, each term in the sum in the first term has expectation equal to $R_\pi(f^*)$ and so, so by weak overlap and bounded second moments of loss, we have $\text{\textcircled{A}} = R_\pi(f^*) + O_p(1/\sqrt{n})$. For term $\text{\textcircled{B}}$, since $h^*$ is fixed we have deterministically that $\text{\textcircled{B}} = O(1/\sqrt{n})$.

Finally, we address term $\text{\textcircled{C}}$, which when expanded can be written as

$$\sup_{\|h\|\le 1} (\frac{1}{m}\sum_{i=1}^m h(\Phi^*(x_i'),t_i') - \frac{1}{n}\sum_{i=1}^n w_i^* h(\Phi^*(x_i),t_i)).$$

Let $x_i'', t_i''$ for $i = 1,\ldots,m$ and $x_i''', t_i'''$ for $i = 1,\ldots,n$ be new iid replicates of $x_1', t_1'$, i.e., new ghost samples drawn from the target design. By Jensen's inequality,

$$\mathbb{E}[\text{\textcircled{C}}^2] = \mathbb{E}[\sup_{\|h\|\le 1} (\frac{1}{m}\sum_{i=1}^m h(\Phi^*(x_i'),t_i') - \frac{1}{n}\sum_{i=1}^n w_i^* h(\Phi^*(x_i),t_i))^2]$$

$$= \mathbb{E}[\sup_{\|h\|\le 1} (\frac{1}{m}\sum_{i=1}^m (h(\Phi^*(x_i'),t_i') - \mathbb{E}[h(\Phi^*(x_i''),t_i'')])$$

$$- \frac{1}{n}\sum_{i=1}^n (w_i^* h(\Phi^*(x_i),t_i) - \mathbb{E}[h(\Phi^*(x_i'''),t_i''')]))^2]$$

$$\le \mathbb{E}[\sup_{\|h\|\le 1} (\frac{1}{m}\sum_{i=1}^m (h(\Phi^*(x_i'),t_i') - h(\Phi^*(x_i''),t_i''))$$

$$- \frac{1}{n}\sum_{i=1}^n (w_i^* h(\Phi^*(x_i),t_i) - h(\Phi^*(x_i'''),t_i''')))^2]$$

$$\le 2\mathbb{E}[\sup_{\|h\|\le 1} (\frac{1}{m}\sum_{i=1}^m (h(\Phi^*(x_i'),t_i') - h(\Phi^*(x_i''),t_i'')))^2]$$

$$+ 2\mathbb{E}[\sup_{\|h\|\le 1} (\frac{1}{n}\sum_{i=1}^n (w_i^* h(\Phi^*(x_i),t_i) - h(\Phi^*(x_i'''),t_i''')))^2]$$

Let $\xi_i(h) = h(\Phi^*(x_i'),t_i') - h(\Phi^*(X_i'^q))$ and let $\zeta_i(h) = w_i^* h(\Phi^*(x_i),t_i) - h(\Phi^*(x_i'''),t_i''')$. Note that for every $h$, $\mathbb{E}[\zeta_i(h)] = \mathbb{E}[\xi_i(h)] = 0$. Moreover, $\mathbb{E}[\|\zeta_i\|^2] \le 4E[K(\Phi^*(x_i'),t_i',\Phi^*(x_i'),t_i')] \le$

$M$. Similarly, $\mathbb{E}[\|\xi_i\|^2] \leq 2E[(w_i^*)^2]M + 2M \leq M' < \infty$ because of weak overlap. Let $\zeta_i'$ for $i = 1, \ldots, n$ be iid replicates of $\zeta_i$ (ghost sample) and let $\epsilon_i$ be iid Rademacher random variables. Because $\mathcal{H}$ is a Hilbert space, we have that $\sup_{\|h\|\leq 1}(A(h))^2 = \|A\|^2 = \langle A, A \rangle$. Therefore, by Jensen's inequality,

$$\mathbb{E}[\sup_{\|h\|\leq 1}(\frac{1}{n}\sum_{i=1}^{n}(w_i^* h(\Phi^*(x_i), t_i) - h(\Phi^*(x_i'''), t_i''')))^2]$$

$$= \mathbb{E}[\sup_{\|h\|\leq 1}(\frac{1}{n}\sum_{i=1}^{n}\zeta_i(h))^2]$$

$$= \mathbb{E}[\sup_{\|h\|\leq 1}(\frac{1}{n}\sum_{i=1}^{n}(\zeta_i(h) - \mathbb{E}[\zeta_i'(h)]))^2]$$

$$\leq \mathbb{E}[\sup_{\|h\|\leq 1}(\frac{1}{n}\sum_{i=1}^{n}(\zeta_i(h) - \zeta_i'(h)))^2]$$

$$= \mathbb{E}[\sup_{\|h\|\leq 1}(\frac{1}{n}\sum_{i=1}^{n}\epsilon_i(\zeta_i(h) - \zeta_i'(h)))^2]$$

$$\leq \frac{4}{n^2}\mathbb{E}[\sup_{\|h\|\leq 1}(\sum_{i=1}^{n}\epsilon_i\zeta_i(h))^2]$$

$$= \frac{4}{n^2}\mathbb{E}[\|\sum_{i=1}^{n}\epsilon_i\zeta_i\|^2]$$

$$= \frac{4}{n^2}\mathbb{E}[\sum_{i,j=1}^{n}\epsilon_i\epsilon_j\langle\zeta_i, \zeta_j\rangle]$$

$$= \frac{4}{n^2}\mathbb{E}[\sum_{i=1}^{n}\|\zeta_i\|^2]$$

$$= \frac{4}{n^2}\sum_{i=1}^{n}\mathbb{E}[\|\zeta_i\|^2]$$

$$\leq \frac{4M'}{n}$$

An analogous argument can be made of $\xi_i$'s, showing that $\mathbb{E}[\textcircled{C}^2] = O(1/n)$ and hence $\textcircled{C} = O(1/\sqrt{n})$ by Markov's inequality. $\qquad\square$

## B  IMPLEMENTATION

We implemented all neural network models (IPM-WNN, RCFR) in TensorFlow as feed-forward fully-connected networks with ELU activations. All architectures have a representation with two hidden layers of 32 and 16 hidden units, and hypotheses (one for each outcome) of 1 layer of 16 hidden units. The networks were trained using stochastic gradient descent with the ADAM optimizer with a learning rate of $10^{-3}$. The batch size was 128. Representations were normalized by dividing by the norm. Weight functions were implemented as 2 hidden layers of 32 units each, as functions of the representation $\Phi$. $\sigma$ in the RBF kernel was set to 1.0. $\lambda_w$ was set to 0.1 and $\lambda_h$ to $10^{-4}$.

## C  EXPERIMENTS

### C.1  SYNTHETIC

We use a two-layer MLP with ELU units and layer sizes 10, 10 as parameterization of the sample weights. Weights are normalized by dividing by the mean.

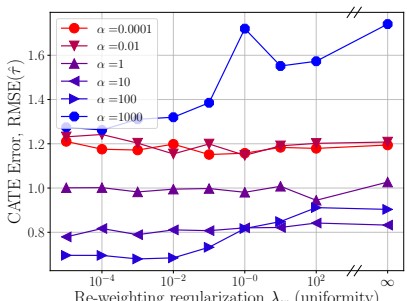 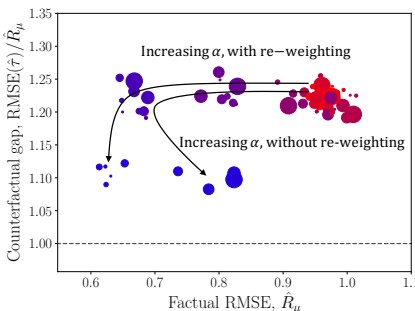

Figure 3: Error in CATE estimation on IHDP as a function of re-weighting regularization strength $\lambda_w$ (left) and source prediction error (right). We see in the left-hand plot that a) for small imbalance penalties $\alpha$, re-weighting (low $\lambda_w$) has no effect, b) for moderate $\alpha$, less uniform re-weighting (smaller $\lambda_w$) improves the error, c) for large $\alpha$, weighting helps, but overall error increases. In the right-hand plot, we compare the ratio of CATE error to source error. Color represents $\alpha$ (see left) and size $\lambda_w$. We see that for large $\alpha$, the source error is more representative of CATE error, but does not improve in absolute value without weighting. Here, $\alpha$ was fixed. Best viewed in color.

## C.2 IHDP

In Figure 3, we see two different views of the IHDP results.

