# OpenReview forum: "Learning Weighted Representations for Generalization Across Designs"
_ICLR.cc/2018/Conference — Reject_

### Official Review · AnonReviewer3 · 2017-11-21
**Reweighting for causal inference in absence of confounding**

**Rating:** 5
**Confidence:** 3

**Review:**

The paper proposes a novel way of causal inference in situations where in causal SEM notation the outcome Y = f(T,X) is a function of a treatment T and covariates X. The goal is to infer the treatment effect E(Y|T=1,X=x) - E(Y|T=0,X=x) for binary treatments at every location x. If the treatment effect can be learned, then forecasts of Y under new policies that assign treatment conditional on X will still "work" and the distribution of X can also change without affecting the accuracy of the predictions.

What is proposed seems to be twofold:
- instead of using a standard inverse probability weighting, the authors construct a bound for the prediction performance under new distributions of X and new policies and learn the weights by optimizing this bound. The goal is to avoid issues that arise if the ratio between source and target densities become very large or small and the weights in a standard approach would become very sparse, thus leading to a small effective sample size.
- as an additional ingredient the authors also propose "representation learning" by mapping x to some representation Phi(x).
The goal is to learn the mapping Phi (and its inverse) and the weighting function simultaneously by optimizing the derived bound on the prediction performance.

Pros:
- The problem is relevant and also appears in similar form in domain adaptation and transfer learning.
- The derived bounds and procedures are interesting and nontrivial, even if there is some overlap with earlier work of Shalit et al.

Cons:
- I am not sure if ICLR is the optimal venue for this manuscript but will leave this decision to others.
- The manuscript is written in a very compact style and I wish some passages would have been explained in more depth and detail. Especially the second half of page 5 is at times very hard to understand as it is so dense.
- The implications of the assumptions in Theorem 1 are not easy to understand, especially relating to the quantities B_\Phi, C^\mathcal{F}_{n,\delta} and D^{\Phi,\mathcal{H}}_\delta. Why would we expect these quantities to be small or bounded? How does that compare to the assumptions needed for standard inverse probability weighting?
- I appreciate that it is difficult to find good test datasets for evaluating causal estimator.  The experiment on the semi-synthetic IHDP dataset is ok, even though there is very little information about its structure in the manuscript (even basic information like number of instances or dimensions seems missing?). The example does not provide much insight into the main ideas and when we would expect the procedure to work more generally.

---

> ### Author Response · Authors · 2018-01-05
> **We respond to the specific concerns of Reviewer 3, both here and in our updated draft.**
>
> Q: The manuscript is written in a very compact style and I wish some passages would have been explained in more depth and detail. Especially the second half of page 5 is at times very hard to understand as it is so dense.
>
> A: We have improved clarity throughout the paper. For page 5 (Theory) specifically, we have adding headings and explanatory comments to provide additional context.
>
> Q: The implications of the assumptions in Theorem 1 are not easy to understand, especially relating to the quantities B_\Phi, C^\mathcal{F}_{n,\delta} and D^{\Phi,\mathcal{H}}_\delta. Why would we expect these quantities to be small or bounded? How does that compare to the assumptions needed for standard inverse probability weighting?
>
> A: We have added comments about the implications of Theorem 1. B_\Phi is determined by the determinant of the Jacobian of the inverse representation \Psi. For smooth invertible representations and an appropriate IPM, we can expect B to be bounded, but it could well be large. As long as we have sufficient overlap, there exists a weighting function to make the IPM term zero, (regardless of the scale of B). C and D are defined explicitly in the appendix and are standard sample complexity terms. Depending on application, Rademacher complexity, VC dimension, or covering numbers could be used for C. This has been clarified. For inverse probability weighting, our bound reduces to that of Cortes et al (2010) as the IPM term vanishes. For other weightings, the added assumption is that the loss lies in the family of functions determining the IPM. A larger family increases the changes of this being true, but loosens the bound in general.
>
> Q: I appreciate that it is difficult to find good test datasets for evaluating causal estimator. The experiment on the semi-synthetic IHDP dataset is ok, even though there is very little information about its structure in the manuscript (even basic information like number of instances or dimensions seems missing?). The example does not provide much insight into the main ideas and when we would expect the procedure to work more generally.
>
> A: The description of IHDP has been improved. We have also added a more targeted synthetic experiment (see above), that confirms our expectation that the usefulness of our method is largest when sample sizes are small. When sample sizes are large, more complex models can be fit and model misspecification can be reduced, thus reducing the usefulness of weighting methods in general. We have added a synthetic experiment in Section 6.1 to demonstrate this further.

---

### Official Review · AnonReviewer1 · 2017-11-24
**Deep architecture for shift invariance in predictive modeling**

**Rating:** 8
**Confidence:** 3

**Review:**

This paper proposes a deep learning architecture for joint learning of feature representation, a target-task mapping function, and a sample re-weighting function. Specifically, the method tries to discover feature representations, which are invariance in different domains, by minimizing the re-weighted empirical risk and distributional shift between designs.
Overall, the paper is well written and organized with good description on the related work, research background, and theoretic proofs.

The main contribution can be the idea of learning a sample re-weighting function, which is highly important in domain shift. However, as stated in the paper, since the causal effect of an intervention T on Y conditioned on X is one of main interests, it is expected to add the related analysis in the experiment section.

---

> ### Author Response · Authors · 2018-01-05
> **We thank Reviewer 1 for their comments.**
>
> We thank Reviewer 1 for their comments.

---

### Official Review · AnonReviewer2 · 2017-11-27
**Good theoretical results, more empirical evaluations can improve the paper**

**Rating:** 7
**Confidence:** 4

**Review:**

Summary:
This paper proposes a new approach to tackle the problem of prediction under
the shift in design, which consists of the shift in policy (conditional
distribution of treatment given features) and the shift in domain (marginal
distribution of features).

Given labeled samples from a source domain and unlabeled samples from a target
domain, this paper proposes to minimize the risk on the target domain by
jointly learning the shift-invariant representation and the re-weighting
function for the induced representations. According to Lemma 1 and its finite
sample version in Theorem 1, the risk on the target domain can be upper bounded
by the combination of 1) the re-weighted empirical risk on the source domain;
and 2) the distributional discrepancy between the re-weighted source domain and
the target domain. These theoretical results justify the objective function
shown in Equation 8.

Experiments on the IHDP dataset demonstrates the advantage of the proposed
approach compared to its competing alternatives.

Comments:
1) This paper is well motivated. For the task of prediction under the shift in
design, shift-invariant representation learning (Shalit 2017) is biased even in
the inifite data limit. On the other hand, although re-weighting methods are
unbiased, they suffer from the drawbacks of high variance and unknown optimal
weights. The proposed approach aims to overcome these drawbacks.

2) The theoretical results justify the optimization procedures presented in
section 5. Experimental results on the IHDP dataset confirm the advantage of
the proposed approach.

3) I have some questions on the details. In order to make sure the second
equality in Equation 2 holds, p_mu (y|x,t) = p_pi (y|x,t) should hold as well.
Is this a standard assumption in the literature?

4) Two drawbacks of previous methods motivate this work, including the bias of
representation learning and the high variance of re-weighting. According to
Lemma 1, the proposed method is unbiased for the optimal weights in the large
data limit. However, is there any theoretical guarantee or empirical evidence
to show the proposed method does not suffer from the drawback of high variance?

5) Experiments on synthetic datasets, where both the shift in policy and the
shift in domain are simulated and therefore can be controlled, would better
demonstrate how the performance of the proposed approach (and thsoe baseline
methods) changes as the degree of design shift varies.

6) Besides IHDP, did the authors run experiments on other real-world datasets,
such as Jobs, Twins, etc?

---

> ### Author Response · Authors · 2018-01-05
> **We respond to the specific concerns of Reviewer 2, both here and in our updated draft.**
>
> Q: In order to make sure the second  equality in Equation 2 holds, p_mu (y|x,t) = p_pi (y|x,t) should hold as well. Is this a standard assumption in the literature?
>
> A: This is a version of the standard, so-called covariate shift assumption (see e.g. Shimodaira, 2000) often made in e.g. domain adaptation. This was referred to only as outcomes being "stationary" in Section 2, but this has been clarified.
>
> Q: Two drawbacks of previous methods motivate this work, including the bias of representation learning and the high variance of re-weighting. According to Lemma 1, the proposed method is unbiased for the optimal weights in the large data limit. However, is there any theoretical guarantee or empirical evidence to show the proposed method does not suffer from the drawback of high variance?
>
> A: The variance of our estimator due to the weighting is accounted for theoretically in our bound by the factor V_\mu and controlled in practice by a penalty on the norm of the weights, see Section 5. A more uniform set of weights yield lower variance but increased bias due to design shift (measured by the IPM term). We have also added a synthetic experiment investigating this, see Section 6.1.
>
> Q: Experiments on synthetic datasets, where both the shift in policy and the shift in domain are simulated and therefore can be controlled, would better demonstrate how the performance of the proposed approach (and those baseline
>  methods) changes as the degree of design shift varies.
>
> A: We have added a small synthetic experiment to highlight the behavior of our model under varying sample sizes, comparing to methods using importance sampling weights. This is complementary to varying design shift.
>
> Q: Besides IHDP, did the authors run experiments on other real-world datasets, such as Jobs, Twins, etc?
>
> A: The Twins experiment, as used by Louizos et al. 2017, was primarily created to evaluate methods for dealing with hidden confounding. This is not the focus of our method as we assume ignorability. We found that in the setting of weak hidden confounding (small proxy noise), the imbalance between “treatment groups” was relatively small, and additional balancing neither hurt nor helped. We did not run experiments on Jobs.

---

### Author Response · Authors · 2018-01-05
**We have taken the comments of the reviewers into account and updated our paper.**

We thank all of the reviewers for their helpful comments and suggestions. Addressing these issues has increased the length of the manuscript, but we are confident that this is justified by the improved quality of the paper. We have responded to the concerns of the reviewers individually below.

---

### Decision · Program_Chairs · 2018-01-29
**ICLR 2018 Conference Acceptance Decision**

**Decision:**

Reject

**Comment:**

The submission provides an interesting way to tackle the so-called distributional shift problem in machine learning. One familiar example is unsupervised domain adaptation. The main contribution of this work is deriving a bound on the generalization error/risk for a target domain as a combo of re-weighted empirical risk on the source domain and some discrepancy between the re-weighted source domain and the target domain. The authors then use this to formulate an objective function.

The reviewers generally liked the paper for its theoretical results, but found the empirical evaluation somewhat lacking, as do I. Especially the unsupervised domain adaptation results are very toy-ish in nature (synthetic data), whereas the literature in this field, cited by the authors, does significantly larger scale experiments. I am unsure as to how much I value I can place in the IHDP results since I am not familiar with the benchmark (and hence my lower confidence in the recommendation).

Finally, I am not very convinced that this is the appropriate venue for this work, despite containing some interesting results.